# Fungal Interactions Strengthen the Diversity-Functioning Relationship of Solid-State Fermentation Systems

Hongxia Zhang,[a,b] Yuwei Tan,[a] Junlin Wei,[a] Hai Du,[a] Yan Xu[a]

[a]Lab of Brewing Microbiology and Applied Enzymology, Key Laboratory of Industrial Biotechnology of Ministry of Education, Synergetic Innovation Center of Food Safety and Nutrition, School of Biotechnology, Jiangnan University, Wuxi, Jiangsu, China
[b]State Key Laboratory of Food Science and Technology, Jiangnan University, Wuxi, China

**ABSTRACT** Traditional fermentation processes are driven by complex fungal microbiomes. However, the exact means by which fungal diversity affects fermentation remains unclear. In this study, we systematically investigated the diversity of a fungal community and its functions during the multibatch *Baijiu* fermentation process. Metabolomics analysis showed that the metabolic profiles of the *Baijiu* were enhanced with an increase in the fermentation time, as determined from the characteristic volatile flavors. High-throughput sequencing technology revealed that the major fungal species involved in sauce-flavor *Baijiu* fermentation are *Pichia* sp. (41.75%, average relative abundance), *Saccharomyces* sp. (13.07%), thermophilic species (9.16%), *Monascus* sp. (6.80%), *Aspergillus* sp. (4.69%), *Schizosaccharomyces* sp. (3.76%), *Thermomyces* sp. (3.74%), and *Zygosaccharomyces* sp. (1.41%). In addition, the fungal diversity increased as the number of fermentation batches increased. Moreover, the increased fungal diversity contributed to the modularity of the fungal communities, wherein *Pichia* sp., *Torulaspora* sp., and *Saccharomyces* sp. maintained the stability of the fungal community. In addition, metatranscriptomics sequencing technologies were used to reconstruct the key metabolic pathways during fermentation, and it was found that the increased microbial diversity significantly promoted glucose-mediated carbon metabolism. Finally, functional gene analysis showed that functional microorganisms, such as *Zygosaccharomyces* and *Pichia*, can enhance fermentation as a result of the high expression of pyruvate decarboxylase and propanol-preferring alcohol dehydrogenase during the metabolism of pyruvate. These results indicate that fungal biodiversity can be exploited to enhance fermentation-based processes via network interactions and metabolism during multiple-batch fermentation.

**IMPORTANCE** Biodiversity and network interactions act simultaneously on the microbial community structure in the *Baijiu* fermentation process, thereby rendering the microbiome dynamics challenging to manage and predict. Understanding the complex fermentation community and its relationship to community functions is therefore important in the context of developing improved fermentation biotechnology systems. Our work demonstrates that multiple-batch fermentation steps increase microbial diversity and promote community stability. Crucially, the enhanced modularity in the microbial network increases the metabolism of flavor compounds and ethanol. This study highlights the power of biodiversity and network interactions in regulating the function of the microbiome in food fermentation ecosystems.

**KEYWORDS** fermentation, microbiome, diversity-functioning relationship, *Baijiu*, fermented foods

Spontaneous fermentation requires the involvement highly diverse fungal communities, and they have a key effect on the quality of fermented foods (1, 2). Although early studies carried out on these systems focused on the description of the

Address correspondence to Hai Du, duhai88@126.com, or Yan Xu, yxu@jiangnan.edu.cn.

The authors declare no conflict of interest.

compositions of the diverse microbial communities, there is currently a growing interest in linking fungal diversity to fermentation functions (3–5). In particular, the diversity of fermented communities gives rise to a complex web of microbial interactions, and previous ecological studies have suggested not only that the interactions between microorganisms shape and drive the microbial community (6) but also that they play a decisive role in the stability and function of the community (7). From the perspectives of microbiology, ecology, and biotechnology, it is therefore important to understand microbial interactions, and such an understanding would also be expected to play a key role in artificially manipulating community combinations and shaping the desired properties of the fermented product.

Sauce-flavor *Baijiu* (a well-known Chinese liquor) is a traditional fermented alcoholic beverage. In 2021, the production capacity of this liquor reached approximately 0.6 million kiloliters, thereby accounting for 8.4% of the Chinese liquor production capacity (i.e., 7.16 million kiloliters). However, the production capacity of sauce-flavor *Baijiu* is limited by the production cycle and liquor yield, of which liquor yield is significantly lower than those of other liquors. In this context, it should be noted that sauce-flavor *Baijiu* is produced by means of repeated batch fermentation, which involves various complex microbiota (1, 8). The production of sauce-flavor *Baijiu* takes place over nine batch processes, namely, two feeding processes (*Xiasha* and *Zaosha* batches) followed by 1 to 7 batches of *Baijiu* production (9). The *Xiasha* batch refers to the beginning of the annual production cycle in October, wherein sorghum is used as the raw material, and half of the total feeding amount is employed. The *Zaosha* batch refers to the second feeding stage, wherein the fermented grain obtained from the *Xiasha* batch is mixed with the second half of sorghum, and all materials are placed in a pit to undergo fermentation. Subsequently, the fermented grain obtained from the first to seventh batch fermentation stages is subjected to repeated processing, as follows. During the repeated batch fermentation process, the fermented grains are recycled after the separation of metabolites (e.g., ethanol and flavor substances) from the fermented grains by distillation. Over the years, it has been proven that the yield and quality of sauce-flavor Baijiu are significantly improved by repeated batch fermentation and, in particular, after three circulation batches (9, 10). This information indicates that the microbial community structure is adjusted by this process and that fermentation is enhanced (1). However, although many microbiological studies have been conducted to reveal the diversity and succession of microbial communities during the fermentation of sauce-flavor *Baijiu*, there is still a gap between understanding the community assemblage and its relation to the function of this microbial ecosystem.

Traditional *Baijiu* fermentation is carried out using mixed cultures (8), and the fungal species involved in the fermentation process have diverse characteristics. For example, the yeast species are the principal microorganisms responsible for alcoholic fermentation (5, 11), during which yeast-yeast interactions occur to determine the acid resistance and flavor composition of the microbial community (12, 13). In addition, filamentous fungi produce enzymes that degrade the raw materials into fermentable sugars (14), supporting the ability of yeast and bacteria to ferment ethanol and produce volatile compounds. Fungal interactions may even occur in the early stages of fermentation and determine the structure and dynamics of the community (15), as well as the metabolism of various flavor substances during the fermentation process (14). The functional changes taking place in the communities of different batches may therefore be affected by both fungal succession and fungal interactions. Based on these considerations, the development of an understanding of the fungal interactions taking place during the repeated batch fermentation of sauce-flavor *Baijiu* should give insight into the basic rules of microbial system assembly and function enhancement, ultimately allowing superior control and improving the fermentation practices during sauce-flavor *Baijiu* production.

Many studies have demonstrated that network analysis is an insightful means to explore microbial interactions between different microbial communities (16). Network

analysis can be used to explore species interactions and microbial network responses under various environmental conditions (17, 18), and networks can identify keystone organisms or functional microbes that may have the greatest effect on a microbial community structure and its potential functions (19, 20). For example, in the soil microbiome, microbial interactions within and between fungal and bacterial communities are important for enhancing the functioning of the ecosystem (21). Significant species diversity is a key feature of natural *Baijiu* microbial communities, and understanding the contribution of this diversity to community function has always been a core issue in industrial *Baijiu* production. It is therefore important to understand the key microbes driving the formation of the functional fermentation group and to determine the metabolic pathways of microbial functional interactions in *Baijiu* fermentation microbiota.

Thus, here, we report the characterization of the microbial communities present in four successive fermentation batches of sauce-flavor *Baijiu*. High-throughput sequencing amplicons were used to assess the diversity of the microbial communities during fermentation, and a network analysis was conducted to explore the microbial interactions associated with microbial dynamics. Furthermore, metatranscriptomic database analysis is used to obtain information regarding the metabolic pathways and core functional contributors. Based on the obtained results, the fermentation mechanism for sauce-flavor *Baijiu* is revealed, which will be expected to help improve *Baijiu* fermentation techniques and enhance the quality of the final product.

## RESULTS

**Fermentation performance of the different batches during fermentation.** The metabolic profiles and ethanol productivity of *Baijiu* are measurable functions of the *Baijiu* fermentation microbial community in terms of its macroscopic performance. Thus, base liquor samples were collected after the fermentation of different batches (16 liquor samples total) to investigate the differences in the volatile flavors using metabolomics methods. As shown in Fig. 1A, a total of 58 metabolites (7 alcohols, 12 organic acids, 27 esters, and 12 other compounds) were identified. Principal-component analysis (PCA) of the volatile flavors showed that the different batches were separated largely in their compositions, and the observed clusters were consistent with the different batches (Fig. 1B). The characteristic volatile flavors obtained during the *Zaosha* brewing process and the first batch were mainly acids (2-methyl-butanoic acid, 3-methyl-butanoic acid, hexanoic acid, pentanoic acid, and 2-methyl-propanoic acid), alcohols (1-butanol and 1-pentanol), and other compounds (Fig. 1C). In contrast, the characteristic volatile flavors of the second and third batches were mainly esters (ethyl 9-hexadecenoate, ethyl oleate, hexadecanoic acid ethyl ester, linoleic acid ethyl ester, octadecanoic acid ethyl ester, and tetradecanoic acid ethyl ester) (Fig. 1C). It should be noted here that the ester profile of *Baijiu* is a key quality metric since the ester components play a key role in determining the flavor of this product. Interestingly, our results showed that the quality of *Baijiu* was enhanced upon increasing the number of fermentation batches, and the ethanol content also increased (see Fig. S1 in the supplemental material). More specifically, the ethanol contents in the four batches were 14.89, 16.87, 20.55, and 22.03 g/kg, wherein the contents of this alcohol in the second and third batches were significantly higher than that in the *Zaosha* batch ($P < 0.01$). This finding also suggests that the function of the microbial community is improved significantly by repeated batch fermentation.

**Microbial diversity and structure during the fermentation of different batches.** High-throughput sequencing was used to determine the microbial community diversity and composition of each fermentation community. Using clean tags to generate operational taxonomic units (OTUs) based on the up-arise clustering method, a total of 14,062 OTUs were generated (average, 12,990). A total of 261 fermented samples were sequenced, and a total of 9,271,025 high-quality sequences were obtained based on internal transcribed spacer 2 (ITS2) gene sequencing, with an average of 34,280 sequences being obtained per sample (see Table S1 in the supplemental material). All fungal sequences were classified into 7,380 OTUs, with an average of 4,814 after

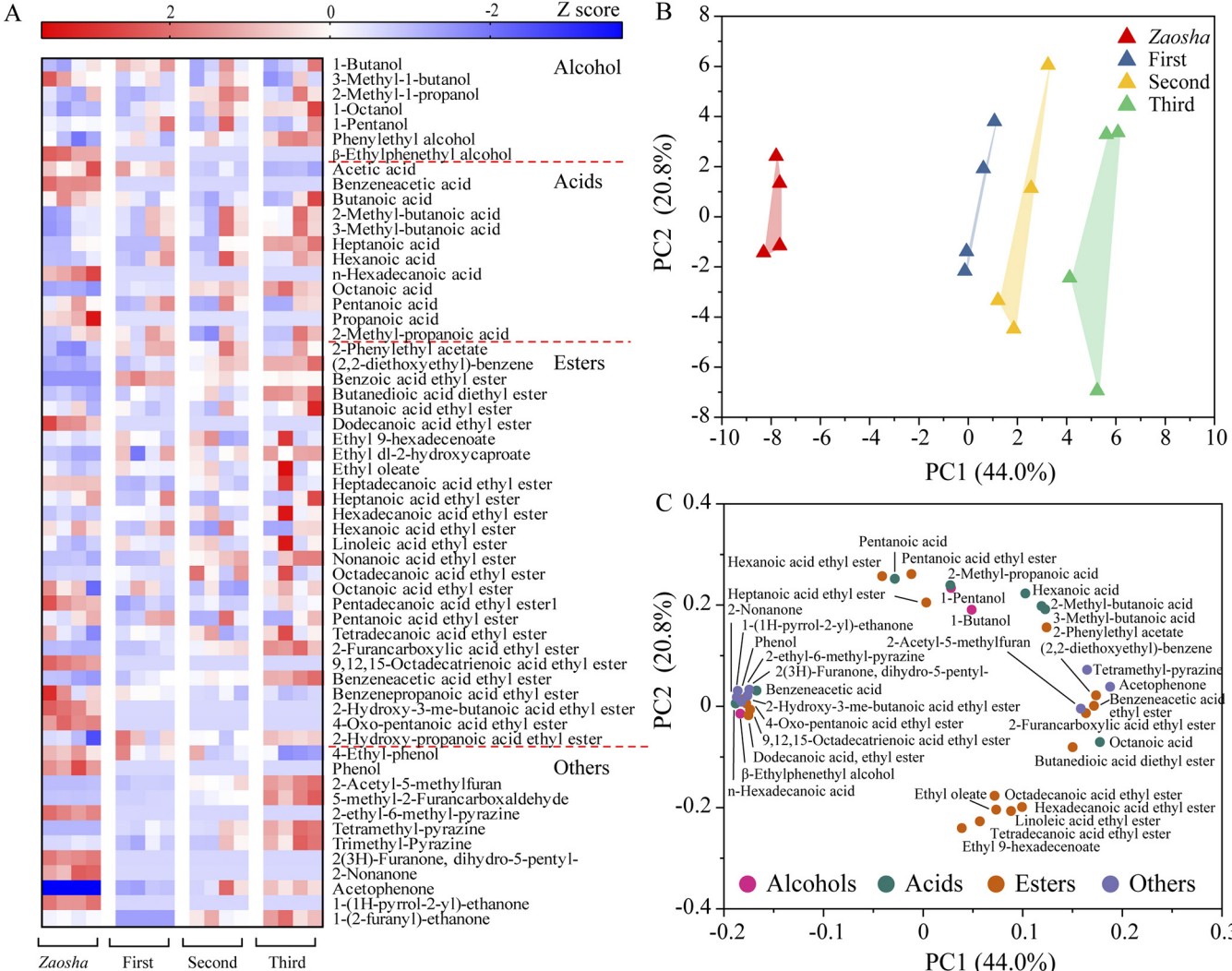

**FIG 1** Metabolic profiles of sauce-flavor *Baijiu* in different batches. (A) Heat map of the metabolites present in *Baijiu* after fermentation. The color scale represents the scaled abundance of each flavor, which is denoted as the *Z*-score, wherein red and blue colors indicate high and low abundances, respectively. (B) Principal-component analysis (PCA) plot of *Baijiu* samples obtained from different fermentation batches. (C) Loading plots generated from the PCA of the metabolic profiles.

flattening. Furthermore, the rarefaction curves, Shannon-Wiener curves (22), and "specaccum" (species accumulation curve) analyses indicated that these sequences were sufficient to reach saturation (see Fig. S2 in the supplemental material).

To elucidate the major species involved in the fermentation process, the relative abundance of each microorganism was investigated at the phylum and genus levels. More specifically, the fungal communities in the fermented grains belonged to 13 different phyla, wherein 3 of these phyla had an average relative abundance of >0.1% (see Fig. S3A in the supplemental material), namely, *Ascomycota* (94.06%), *Basidiomycota* (1.34%), and *Mucoromycota* (0.18%). At the genus level, 88 genera were detected, and 8 genera exhibited average relative abundances of >1% (Fig. 2A), namely, *Pichia* (41.75%), *Saccharomyces* (13.07%), thermophilic species (9.16%), *Monascus* (6.80%), *Aspergillus* (4.69%), *Schizosaccharomyces* (3.76%), *Thermomyces* (3.74%), and *Zygosaccharomyces* (1.41%). Notably, during the fermentation of the *Zaosha* batch, the average relative abundance of *Pichia* reached 76%.

To understand the fungal diversity, the $\alpha$ and $\beta$ diversities were analyzed. The fungal $\alpha$ diversity (OTU richness) increased with an increase in the number of fermentation batches (Fig. S3B) and reached its highest value in the second batch. Partial least-squares (PLS)-discriminant analysis (DA) showed significant differences in the $\beta$ diversity, and the

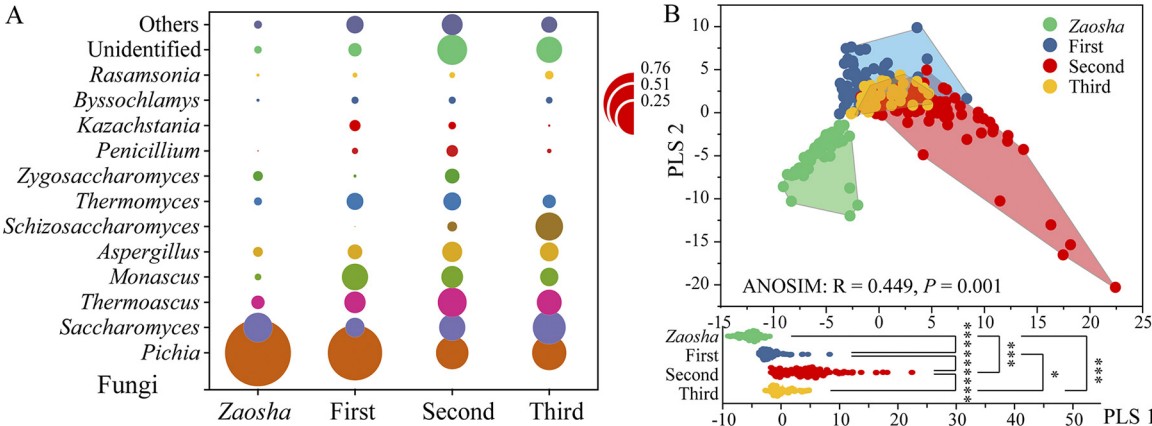

**FIG 2** Fungal community structures of the fermented grains in different batches. (A) The structure of the fungal community at the genus level was significantly different in the different fermentation batches. (B) Partial least-squares discriminant analysis (PLS-DA) of the fungal communities in the different batches.

first principal component differentiated between the fermentation batches (Fig. 2B). This result indicates that the significant differences in the fungal community structures of the different batches were caused mainly by the differences between the fermentation batches. Thus, the diversity analyses indicated that repeated fermentation could lead to enhanced species richness and changes the community structure during the fermentation process.

**Insight into the interactions and keystone taxa across the different batches.** To understand the microbial interactions taking place in the different fermentation batches, a random matrix theory-based network approach was used to reconstruct the fungal networks, in which the species (nodes) were connected by pairwise interactions (links). The topological properties of the fungal networks are summarized in Table 1. As indicated, the same threshold (Similarity threshold, $S_t$ = 0.70) was selected for all fungal networks. In addition, for the whole fermentation community, the network topology fit well with the power law distribution ($R^2$ = 0.92) and showed scale-free properties, thereby indicating that a small number of microorganisms in the network possess a large number of connections, while the majority of microorganisms possess only a few connections (23). Furthermore, the whole fungal network was found to exhibit clear general network properties, such as small world (harmonic geodesic distance = 4.35) and modularity ($M$ = 0.64) properties. Subsequently, to determine the network succession across different batches, individual networks were constructed for each fermentation batch. It was found that the $M$ values in the empirical networks were higher than those from the corresponding randomized networks, thereby indicating that these fungal networks appeared to be modular, wherein the modularity of the network increased with the number of fermentation batches (Fig. 3A). In addition, the average path distance was short (<5) for each batch, indicating that the fermentation communities can quickly adjust to the effects of multiple fermentations. Overall, the above

**TABLE 1** Topological properties of the fungal molecular ecological networks during multibatch fermentation[a]

| Dataset | Empirical networks | | | | | | | Random networks | | |
| --- | --- | --- | --- | --- | --- | --- | --- | --- | --- | --- |
| | Network size (*n*) | Total no. of links | $R^2$ of power law | avgK | avgCC | HD | M | avgCC | HD | M |
| All | 837 | 1,785 | 0.92 | 4.27 | 0.17 | 4.35 | 0.64 | 0.01 ± 0.00 | 3.82 ± 0.02 | 0.48 ± 0.00 |
| *Zaosha* | 143 | 345 | 0.86 | 4.83 | 0.18 | 3.03 | 0.45 | 0.08 ± 0.01 | 2.71 ± 0.04 | 0.38 ± 0.01 |
| First | 191 | 282 | 0.90 | 2.95 | 0.17 | 2.96 | 0.72 | 0.02 ± 0.01 | 3.57 ± 0.07 | 0.59 ± 0.01 |
| Second | 206 | 236 | 0.92 | 2.29 | 0.12 | 5.02 | 0.85 | 0.01 ± 0.01 | 4.36 ± 0.15 | 0.72 ± 0.01 |
| Third | 117 | 106 | 0.94 | 1.81 | 0.11 | 3.27 | 0.85 | 0.01 ± 0.01 | 4.46 ± 0.39 | 0.81 ± 0.01 |

[a]avgK, avg degree; avgCC, avg clustering coefficient; HD, harmonic geodesic distance; M, modularity.

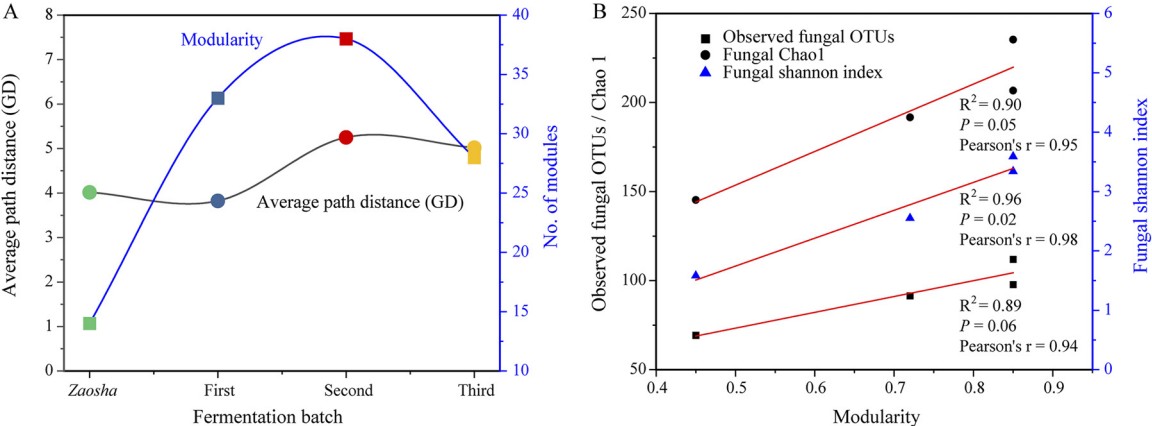

**FIG 3** Variation in the fungal network features in different fermentation batches. (A) Average path distances (GDs) and modularities in different batches. (B) Relationship between the microbial diversity and the community modularity.

observations show that an increased fungal diversity contributed to the modularity of the community (Fig. 3B).

As shown in Fig. S4 in the supplemental material, the networks changed dynamically from being complex in the *Zaosha* and first batches to being simple in the second and third batches. A total of 969 edges were identified, and positive associations between the taxa (66.98% to 97.10%, cf., 2.90% to 33.02% negative associations) were found to dominate all fermentation batches. However, the percentage of negative correlations increased notably with the number of fermentation batches (i.e., 2.90%, 17.38%, 32.20%, and 33.02%, respectively) (see Table S2 in the supplemental material), which could reflect competition for limited resources or distinctive environmental niches produced by microbial metabolism. In addition, the network of the second batch appeared to have sparser connections between network members (Table 1), which was reflected by the large harmonic geodesic distance (HD), i.e., the smallest value of the average clustering coefficient (avgCC). In addition, the second and third batches showed a short average degree (avgK; 2.29 and 1.81, respectively) and the highest modularity (both, 0.85), which means that the fungal community can communicate more effectively and adapt rapidly to the fermentation environment in the later batches. Thus, multiple-batch fermentation appears to reduce the network complexity of the microbial community and improve the symbiotic relationships and network stability.

Since the altered network complexity could lead to changes in the roles of individual member species within the network, two parameters (i.e., the within-module connectivity [$Zi$] and the module connectivity [$Pi$]) were used to decipher the topological roles of the OTUs in the networks. A total of 188 connectors (nodes linking different modules) and 6 network hubs (nodes being both module hubs and connectors) were detected across the molecular ecological networks (MENs) (Fig. 4). The six network hubs included *Torulaspora delbrueckii*, *Pichia membranifaciens*, *Hypocreales* sp., *Annulohypoxylon stygium*, *Talaromyces rugulosus*, and *Saccharomyces mikatae*. These microorganisms are highly connected to many other nodes within their modules and are thereby regarded as keystone nodes that play key roles in shaping the network structure. Additional keystone nodes were also observed in the second batch, and 2 network hubs and 32 connectors were identified. These key microorganisms are classed as filamentous fungi, including *Thermoascus aurantiacus*, *Thermoascus crustaceus*, *Thermomyces lanuginosus*, *Leiothecium ellipsoideum*, *Microascus brevicaulis*, *Aspergillus costiformis*, *Rhizopus microspores*, and *Monascus purpureus*. Many of the keystone node-affiliated taxonomic groups appeared to be important in the degradation of raw materials (starch and protein). In particular, the *Thermoascus* sp. keystone nodes were preserved across multiple batches, indicating that in multibatch fermentation, the MEN is conservative at a high tissue level. Keystone fungi therefore not

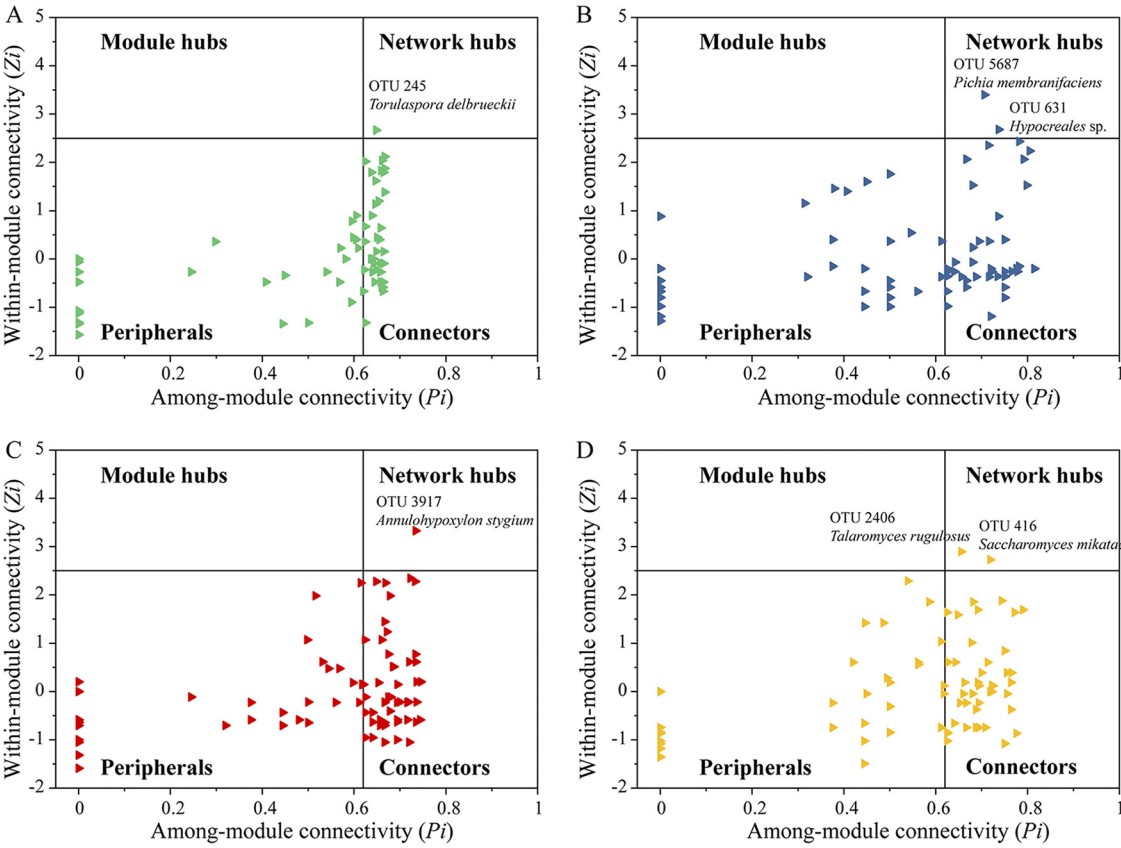

**FIG 4** Putative keystone taxa identified based on the node topological roles in the networks of different batches (A, *Zaosha* batch; B, first batch; C, second batch; D, third batch). Each symbol represents a node in one of the networks. A node was identified as a module hub if its within-module connectivity ($Zi$) was ≥2.5, as a connector if its module connectivity ($Pi$) was ≥0.62, and as a network hub where $Zi$ is ≥2.5 and $Pi$ is ≥0.62.

only play a key role in promoting the fermentation process but also are responsible for increasing the community stability.

**Fungal interactions strengthen the diversity-functioning relationship.** An intriguing and important question is how the diversity and network complexity affect the microbial community function during multiple-batch fermentation processes. To answer this question, we initially correlated the relationships among the modularity, ethanol content, total acidity, and total ester content based on Pearson's correlations. As indicated in Fig. 5A and B, the ethanol and ester contents were positively correlated with the modularity in fungal communities ($P = 0.083$ and $P = 0.136$, respectively), while there was a significant negative correlation between the total acidity and the modularity ($P < 0.05$). These results indicated that upon increasing the number of fermentation batches, the enhancement in the modularity was beneficial for the generation of ethanol and esters but inhibited the accumulation of acids. Moreover, to explore the changes taking place in the functional genes and the metabolic pathways, metatranscriptomic samples collected from the *Zaosha* ($n = 4$) and second batches ($n = 4$) were analyzed using metatranscriptomic analysis. Using the *de novo* assembly program Trinity, a total of 222,341 contigs were assembled. For annotation, similarity searches were performed to annotate the unigenes against different databases using BLASTX. As a result, we observed the same dominant genera contributing to our metagenomic data set, including *Pichia*, *Zygosaccharomyces*, *Schizosaccharomyces*, and *Saccharomyces*, which indicates that the dominant taxa in our data set were transcriptionally active.

To reveal the metabolic differences between the microorganisms of different fermentation batches, the gene expression patterns in the different batches were analyzed.

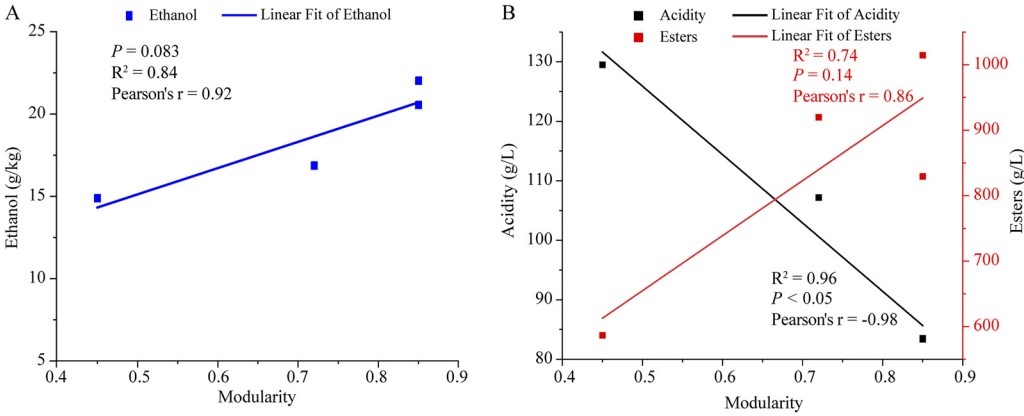

**FIG 5** The relationships between the modularity in the network and the fermentation performance. (A) The relationship between the modularity and the ethanol content based on Pearson's correlations. (B) The relationships between the modularity and the total acidity and total esters.

Initially, PCA revealed altered community transcriptional activity in the fermented grains, with significant clustering by batch (Fig. 6A; analysis of similarity [ANOSIM] test, $R = 0.9583$, $P = 0.027$). Subsequently, differential micro-RNA (miRNA) expression analysis of the two libraries was performed using the DEGseq R package. Significant differentially expressed genes (DEGs) were evaluated against a $|\log_2 (\text{fold change})|$ of $\geq 2$ and a false discovery rate (FDR) of $<0.05$. It was found that 7,901 genes were significantly differentially expressed, including 5,176 upregulated genes and 2,725 downregulated genes (Fig. 6B). Moreover, using the gene ontology (GO) annotation, the possible functions of the DEGs were classified (Fig. 6C and D). Thus, the main upregulated genes in the second batch were found to be distributed across different biological processes (including organonitrogen compound metabolic, organonitrogen compound biosynthetic, cellular protein metabolic, carboxylic acid metabolic, and oxidation-reduction processes), cellular components (including organelles, intracellular organelles, membrane-bound organelles, and intracellular membrane-bounded organelles), and molecular functions (structural molecule activity and structural constituent of ribosome). Furthermore, KEGG pathway enrichment analysis involving the DEGs was conducted by mapping to the KEGG Pathway database (see Fig. S5 in the supplemental material). As a result, the enriched factors of the upregulated DEGs were related to protein export, pentose and glucuronate interconversion, galactose metabolism, vitamin B6 metabolism, and autophagy. Similarly, the enriched factors of the downregulated DEGs were involved in butanoate metabolism; nitrogen metabolism; the alanine, aspartate, and glutamate metabolism; and the glycine, serine, and threonine metabolism. These results indicate that multibatch fermentation significantly improves the functionality of the second batch process and that the microbial metabolism taking place during substrate degradation and during glucose and pyruvate biosynthesis may have important roles in both flavor development and alcohol accumulation during the fermentation process.

Combined with the above DEGs and associated pathways, metabolic network analysis was performed (Fig. 7). A total of 16 genes were annotated for starch catabolism, and the expression level in the second batch was 5.37 times greater than that in the *Zaosha* batch (Fig. 7A and D, Fig. 8A). In addition, glucose catabolism was significantly enhanced in the second fermentation round and was 16.91 times greater than the expression level in the *Zaosha* batch (Fig. 7B and D). Furthermore, pyruvate metabolism became increasingly active as a major metabolic pathway in the second batch; in particular, the metabolism of ethanol and acetic acid was enhanced, which is of particular importance in terms of the metabolism of alcohol and flavor compounds (Fig. 7C and D, Fig. 8C).

To explain the correlations between the core microorganisms and the major enriched genes, the key genes that could play a regulatory role in the metabolism

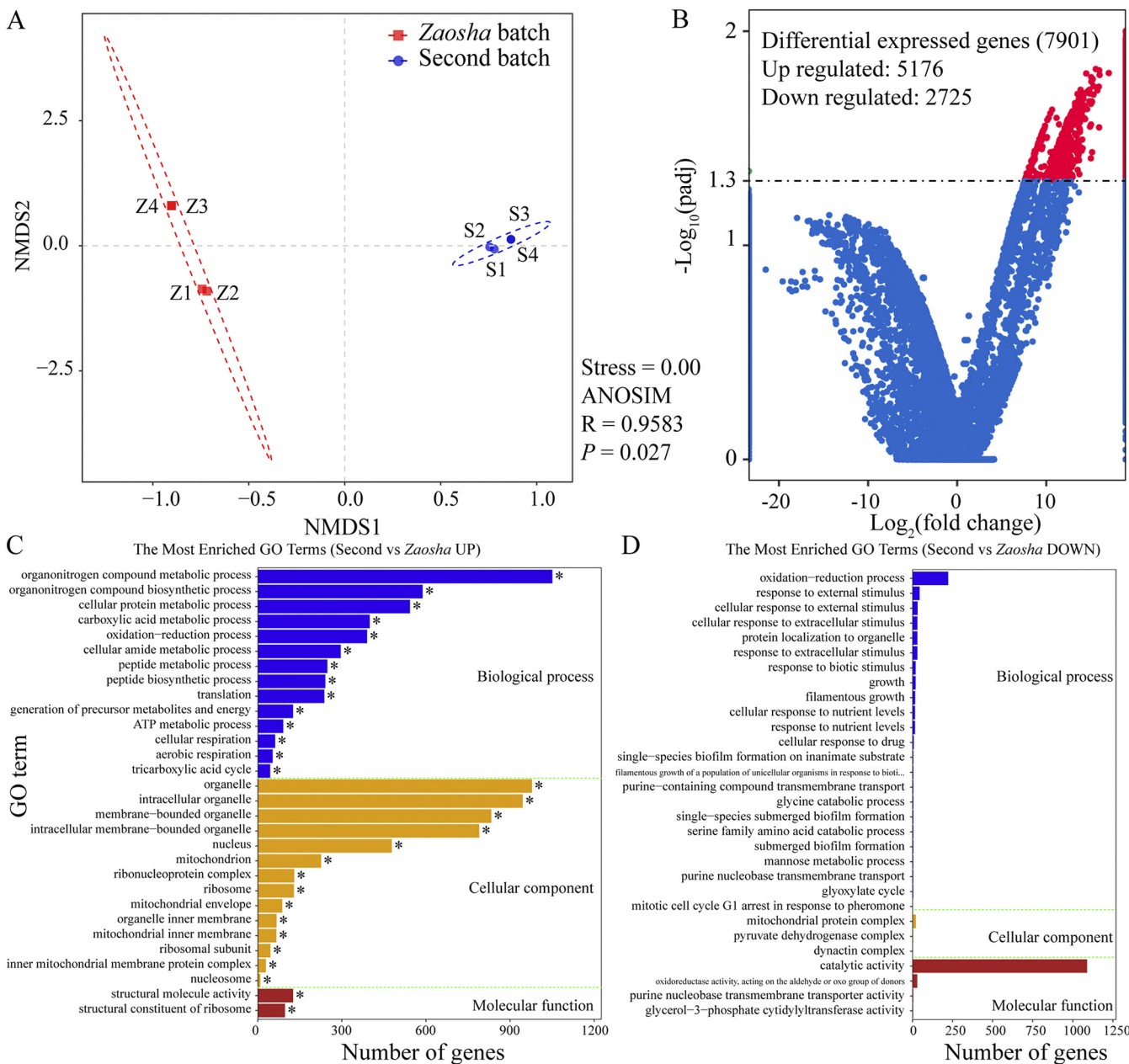

**FIG 6** The differentially expressed genes (DEGs) in *Zaosha* versus the second batch. (A) Nonmetric multidimensional scaling (NMDS) plot of the samples based on transcriptome analysis. (B) Volcano plots showing the number of differentially expressed genes. The DEGs are shown in red (upregulated) and blue (downregulated), whereas the gray color indicates genes that were not differentially expressed. (C, D) Gene ontology (GO) analysis of differentially expressed genes.

process were further analyzed. Although the functional genes were expressed mainly by *Pichia*, *Ogataea*, and *Brettanomyces* in the *Zaosha* batch, their expression levels were low (Fig. 8). In contrast, functional genes related to the genera *Zygosaccharomyces*, *Pichia*, *Schizosaccharomyces*, *Saccharomyces*, and *Torulaspora* were expressed at high levels in the second batch. During the metabolism of starch and sucrose (Fig. 8A), a variety of yeast and filamentous fungi participated in the gene expression of 1,3-$\beta$-glucan synthase (KEGG database, K00706), glucan-1,3-$\beta$-glucosidase (K01210), and 6-phosphofructokinase 1 (pfkA, PFK; K00844). These highly expressed pathways and genes greatly accelerated the production and utilization rates of glucose and fructose. In addition, it was found that the expression of glycolysis/gluconeogenesis genes, including glyceraldehyde 3-phosphate dehydrogenase (GAPDH, gapA; K00134); fructose-bisphosphate aldolase, class II

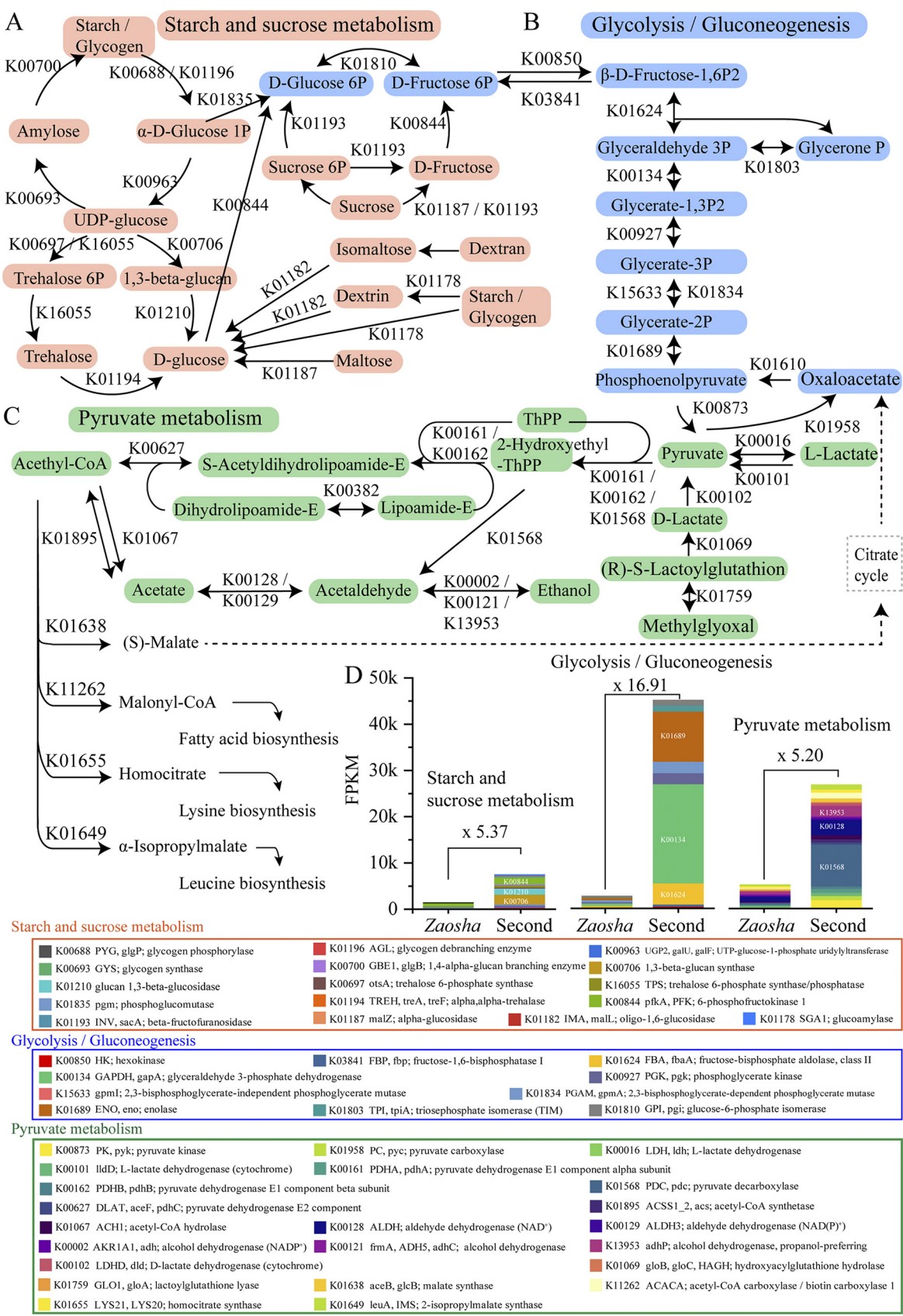

**FIG 7** Metabolic network for alcohol fermentation of sauce-flavor *Baijiu* production. (A) Starch and sucrose metabolism. (B) Glycolysis/gluconeogenesis. (C) Pyruvate metabolism. (D) Fragments per kilobase of transcript per million (FPKM) of the KEGG expression genes related to the three main metabolic networks during the fermentation process.

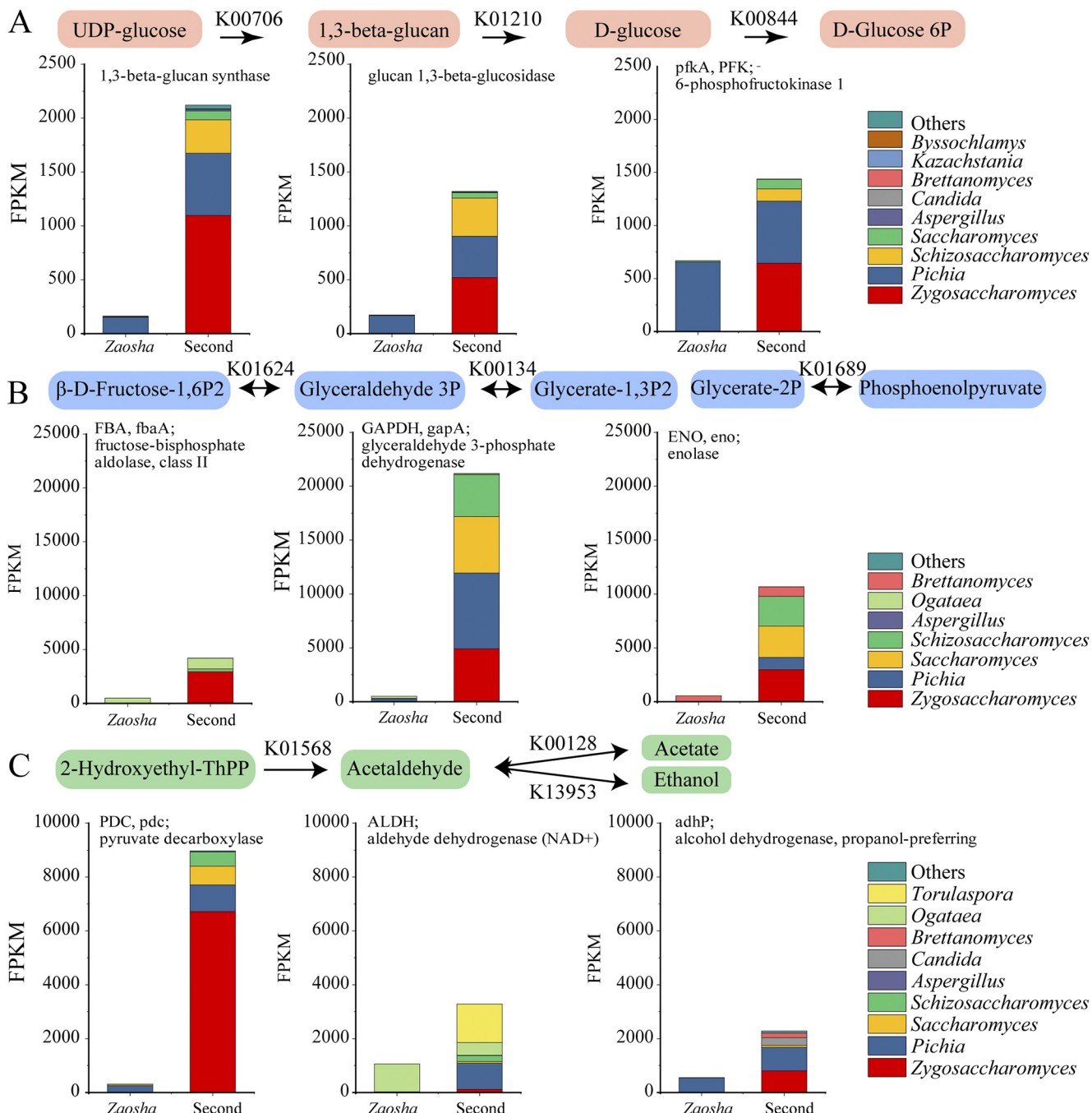

**FIG 8** Microbial metabolic activity during sauce-aroma *Baijiu* fermentation. (A) The top three enriched fungal genes expressed during starch and sucrose metabolism. (B) The top three enriched fungal genes expressed during glycolysis/gluconeogenesis. (C) The top three enriched fungal genes expressed during pyruvate metabolism.

(FBA, fbaA; K01624); and enolase (ENO, eno; K01689), was significantly increased by 41.56, 8.60, and 19.41 times compared with that in the *Zaosha* batch (Fig. 7D and 8B). Furthermore, six yeasts, including *Zygosaccharomyces*, *Pichia*, *Saccharomyces*, *Schizosaccharomyces*, *Ogataea*, and *Brettanomyces*, were highly expressed (Fig. 8B), indicating that the increased microbial diversity significantly promoted the glucose-mediated carbon metabolic pathway. In addition, pyruvate decarboxylase (PDC) from *Zygosaccharomyces* was significantly overexpressed compared with that of other yeasts; this enzyme catalyzes the conversion of pyruvate to acetaldehyde and serves as a

precursor for the metabolism of acetate and ethanol. Moreover, genes related to the genera *Pichia*, *Zygosaccharomyces*, *Candida*, *Brettanomyces*, *Saccharomyces*, *Kluyveromyces*, *Talaromyces*, and *Wickerhamomyces* showed high metabolic capacities (e.g., propanol-preferring alcohol dehydrogenase [adhP; K13953]) for the conversion of ethanol to acetaldehyde in the fermentation process. Based on the expression levels of the functional microorganisms, *Pichia* was identified as the main active microorganism in the *Zaosha* batch, while *Zygosaccharomyces* was expressed at higher levels in the second batch (see Fig. S6A in the supplemental material). The other yeast species mentioned above were essentially absent from the *Zaosha* batch, but their presence significantly increased in the second batch (Fig. S6).

## DISCUSSION

*Baijiu* fermentation involves a large and diverse contribution of different microbes (24–26) and results in a complex microbial community that has a direct effect on fermentation. In this study, the fungal microbial community diversity and composition were characterized for different fermentation batches, and repeated batch fermentation was found to increase the community diversity. Interestingly, this increased diversity improved the interspecific interactions and network stability, and identification of the core microorganisms in each community provided a possible strategy for the steady-state regulation of the community. In addition, the diversity-functioning relationship was revealed by transcriptome analysis, and it was determined that the fungal interactions strengthened the metabolism of starch and sucrose during fermentation, in addition to promoting glycolysis/gluconeogenesis and enhancing the metabolism of pyruvate. In terms of the production of the sauce-flavor *Baijiu* fermentation process, multibatch fermentation promoted community function by increasing its diversity and network stability. These results demonstrate that more stable microbial networks contribute to an improved multifunctional fermentation process than those achieved using simpler or low-diversity networks.

Multibatch fermentation appears to regulate community function under natural conditions, and significant differences in the metabolic profiles were observed between the different fermentation batches. As reported previously, esters, acids, and alcohols are important volatile compounds in sauce-flavor *Baijiu*, and so their compositions and concentrations greatly influence the quality index of the final product (10). More specifically, esters are essential ingredients that yield fruity and floral flavors, and they are an important group of volatile compounds that contribute to the flavor of *Baijiu* (27). The significantly increased ester composition and content in the second batch therefore indicate that multibatch fermentation improves the quality of *Baijiu* (Fig. 1). Acids are also an important group of chemicals in the fermentation of *Baijiu* because they can greatly suppress the growth of spoilage-related microorganisms, in addition to being important substrates for ester synthesis (8, 27). In the *Zaosha* and first batches, the contents of acetic acid, propionic acid, and butyric acid were high, and these components were expected to act as precursor substances for later batches. In the second and third batches, the accumulation of alcohols was detected, which are key substrates for the preparation of esters and other important flavor chemicals in *Baijiu*. The ester compounds present in *Baijiu* are generated mainly by microbes (27), with the yeasts being the main producers (28, 29). In contrast, filamentous fungi, such as *Monascus* and *Rhizopus*, secrete large amounts of esterase (30, 31), and so this enriched microbial diversity across different batches ultimately has a significant effect on the anabolism of the different flavor substances.

The enhanced microbial diversity also affects the complexities and stabilities of the various ecological networks in the microbial communities during *Baijiu* fermentation. Indeed, recent studies have identified the diversity of the microbiome complexity and composition in fermentation processes (8, 14, 32). For example, microorganisms in the phylum Ascomycota, including yeast and filamentous fungi, are the main fungal populations in a variety of fermented foods (2, 26, 33). In our study, significant differences in

the structure of the microbial community were observed between the different fermentation batches. More specifically, *Pichia* were found to be the dominant microbial population in the sauce-flavor *Baijiu* fermentation process (Fig. 2), in addition to being the main functional microorganisms of various types of liquor (34–36), which indicates their key role in the fermentation microecology (1, 8, 15). From a network perspective, positive associations dominated all fermentation batches (Table S2) and suggested mostly cooperative behaviors, such as syntrophic interactions and shared environmental requirements. In the later fermentation steps, the short average degree and high modularity of the network meant that the community was able to communicate effectively and adapt rapidly to the effects and changes in the fermentation environment. Interestingly, the increased diversity reduced the network complexity (Table 1) but improved the modularity and stability of the microbial community (Fig. 3). Furthermore, the keystone node-affiliated taxonomic groups appeared to be important in the context of substrate degradation, as well as in the adaptation to elevated temperatures. These results show that filamentous fungi play a key role not only in promoting the fermentation process but also in stabilizing the microbial community.

Furthermore, we found that microbial interactions are conducive to the improvement of community function. The first step in the *Baijiu* fermentation process is the metabolism of starch and sucrose, and we found that an improved microbial diversity enhanced the expression of 1,3-$\beta$-glucan synthase and glucan 1,3-$\beta$-glucosidase, in addition to accelerating the conversion of UDP-glucose to D-glucose (Fig. 7 and 8). Previous studies have shown that the variations in the starch and fermentable sugar contents were consistent with the growth of filamentous fungi (such as *Paecilomyces variotii* and *Aspergillus oryzae*) in fermented grains during sauce-flavor *Baijiu* fermentation, which implied that filamentous fungi play an important role in producing amylase for the hydrolysis of starch (14). Conversely, an increase in the fermentable sugar content indicated an increased activity in terms of the fungal metabolism, and a greater fungal diversity also had a positive effect on the decomposition and utilization of reducing sugars and metabolite synthesis. Using glucose as a substrate, microorganisms can produce ethanol and metabolize flavor compounds through the process of glycolysis, during which PFK is a key rate-limiting enzyme that plays a critical role in the glycolytic pathway. To maintain a high level of PFK expression, the functional cooperation exhibited by the genera *Zygosaccharomyces*, *Pichia*, *Schizosaccharomyces*, and *Saccharomyces* was essential. Such functional cooperation is also reflected in the supplementation of various metabolic pathways. For example, the metabolism of glyceraldehyde 3-phosphate (3P) has two main functions, namely, energy generation and precursor synthesis. More specifically, *Zygosaccharomyces* and *Ogataea* mainly express fructose-bisphosphate aldolase class II (FBA, fbaA) and catalyze the transformation of $\beta$-D-fructose-1,6P2 to glyceraldehyde 3P in the first batch fermentation. In addition, the production of glycerate-1,3P2 by glyceraldehyde 3P under the action of glyceraldehyde 3-phosphate dehydrogenase (GAPDH, gapA) is carried out mainly by four yeast species, namely, *Pichia*, *Saccharomyces*, *Zygosaccharomyces*, and *Schizosaccharomyces* (Fig. 8). During the metabolism of pyruvate, the high expression of pdc by *Zygosaccharomyces* ensures a supply of precursors for the metabolism of ethanol and acetate. The high expression of adhP also ensures that ethanol is produced in large quantities during the second batch fermentation, complements the conversion of pyruvate to ethanol via acetyl-coenzyme A (CoA), acetate, and acetaldehyde (5). In addition, the high level of expression of the alcohol dehydrogenase gene was found to be consistent with the increased alcohol content during the second batch fermentation (Fig. S1). More importantly, the enhanced pyruvic acid metabolic pathway simultaneously promotes the metabolism of fatty acids, amino acids, and other compounds, which promote the diversification of the final flavor profile. Our results therefore suggest that manipulating the fungal diversity is a novel strategy for improving the community stability and function in a spontaneous fermentation system. Furthermore,

these results may aid in the development of a controlled fermentation process to improve the quality, productivity, and safety of fermented foods.

In conclusion, understanding the interactions between fungal taxa and their relationships with metabolic profiles has important implications in spontaneous fermentation. In our study, the metabolic profiles, interactions, and metabolic pathways of fungal communities across four fermentation batches were studied systematically. We concluded that the metabolic profiles were improved with an increase in the number of fermentation batches. In particular, community diversity showed an obvious increasing trend when a greater number of fermentation batches were employed, and multibatch fermentation was found to enhance the activity of the functional microorganisms. The communities consistently displayed network properties in different fermentation batches; more specifically, the network structure became simpler, and the modularity of the network increased. In this context, *Torulaspora delbrueckii*, *Pichia membranifaciens*, *Hypocreales* sp., *Annulohypoxylon stygium*, *Talaromyces rugulosus*, and *Saccharomyces mikatae* were proposed to play essential roles in network stabilization. Furthermore, transcriptome analysis further revealed the effects of community stability on the metabolic pathways for the metabolism of starch and sucrose, the glycolysis process, and the metabolism of pyruvate. In general, our study provides insight into how fungal communities respond to multiple-batch fermentation; however, further studies on bacteria and the metabolic processes related to more complex flavor compounds are required. Moreover, to improve the productivity of sauce-flavor *Baijiu*, besides identifying core microbial communities, an in-depth understanding of microbial interactions is also critical.

## MATERIALS AND METHODS

**Sampling.** We collected samples during the fermentation process (heap and pit fermentation) in four batches in 2018 and 2019 in a *Baijiu* distillery of sauce-flavor liquor in Guizhou Province, China (28.14 N, 106.18 E). Samples taken from different points in the same layer were mixed to form a single sample to reduce the heterogeneity before extraction and analysis (1, 8). We tracked the fermentation process of four batches, and the number of samples was determined according to the fermentation time; also, at least two pits were selected for each round as the research object. In 2 years, we collected 261 fermented samples, including 65 samples of *Zaosha* batch, 62 samples of first batch, 90 samples of second batch, and 44 samples of third batch. The base *Baijiu* for volatile compounds analysis included 16 samples, of which representative samples were taken at different 4 batches. We evaluated and compared the proportions and changing trends of alcohols, acids, esters, and other heterocyclic compounds in base *Baijiu*, indicating the differences in community functions between fermentation batches. In addition, to compare the microbial community function and related metabolic processes in the fermentation process, we chose the samples from *Zaosha* and second batches for transcriptome sequencing ($n = 4$).

**Volatile flavor compound analysis.** Fermented grains (5 g) were added to 20 mL sterile saline (0.85% NaCl and 1% CaCl$_2$), ultrasonically treated for 30 min (in an ice-water mixture as 0°C), and then centrifuged at 8,000 $\times$ $g$ for 5 min (4°C). Eight milliliters of supernatant and 10 $\mu$L menthol (internal standard, 100 $\mu$g/mL) were placed into a 20-mL headspace vial with 3 g NaCl (37).

The volatile compounds of base *Baijiu* were analyzed using headspace solid-phase microextraction coupled to gas chromatography-mass spectrometry (HS-SPME-GC-MS; GC 6890N and MS 5975; Agilent Technologies, Santa Clara, CA) with a DB-Wax column (30-m by 0.25-mm internal diameter [i.d.], 0.25-$\mu$m film thickness; J&W Scientific, Folsom, CA), based on the method described for a previous study (38). Briefly, helium at a constant flow rate of 2 mL/min was used as a carrier gas. The injector was held at 250°C. The oven temperature was held at 50°C for 2 min, then programmed to 230°C at a rate of 6°C/min, and finally kept at 230°C for 20 min. The column flow was split at the end of the capillary; one was directed to a heated olfactometer (Olfactory Detector Port ODP 2; Gerstel Inc., Mülheim, Ruhr, Germany), whereas the other one was directed to the MSD (Mass spectrometer detector). The temperature of the olfactory port was 280°C. Mass spectra in the electron ionization mode (MS-EI) were recorded at 70-eV ionization energy, and the ion source temperature was set at 230°C. Full-scan acquisition was used in the 30- to 350-atomic mass unit (amu) range of masses.

A volatile compound was considered to be identified tentatively if the similarity between mass spectrometric information of each chromatographic peak and the National Institute of Standards and Technology (NIST) mass spectra library was at least 75% and the difference between RIcal and RIlit did not exceed 30 units (38, 39). Identified volatile compounds were semiquantified by comparison with the peak areas of the internal standard.

**DNA and RNA extraction and sequencing.** All samples were treated with sterile phosphate-buffered saline (0.1 mol/L) and then centrifuged at 300 $\times$ $g$ for 10 min (4°C). The supernatant was then centrifuged again at 13,000 $\times$ $g$ for 10 min (4°C) to precipitate cells.

For DNA extraction and sequencing, the precipitated cells were milled with liquid nitrogen (5), and genomic DNA was extracted using a sodium laurate buffer (sodium laurate, 10 g/L; Tris-HCl, 0.1 mol/L; NaCl, 0.1 mol/L; EDTA, 0.02 mol/L) containing phenol-chloroform-isoamyl alcohol (25:24:1, vol/vol/vol).

PCR products were purified using a PCR purification kit, and the concentrations were assessed with a Thermo Scientific NanoDrop 8000 UV-visible (UV-Vis) spectrophotometer (NanoDrop Technologies, Wilmington, DE). The barcoded PCR products were sequenced on a MiSeq benchtop sequencer (Illumina, San Diego, CA) for 250-bp paired-end sequencing at Beijing Auwigene Tech, Ltd. (Beijing, China). The internal transcribed spacer 2 (ITS2) region was amplified with primers ITS3 (5′-GCA TCG ATG AAG AAC GCA GC-3′) and ITS4 (5′-TCC TCC GCT TAT TGA TAT GC-3′) (40). All the raw sequences generated were processed via QIIME v.1.9.1 (41) and R v.3.3.1 (http://www.r-project.org). The representative fungal OTU sequences were compared using a BLAST search against the UNITE fungal ITS database (https://unite.ut.ee/). The data of each sample were subjected to homogenization treatment, and the subsequent diversity analysis was based on the homogenized data. The alpha diversity analysis was made using the OTU table by QIIME software (v.1.9.1). Multivariate statistical analyses on microbial community were performed using the OTU table with the SIMCA-14.1 software (Umetricus, Sweden).

For RNA extraction and sequencing, the precipitated cells were collected according to the method of Song et al. (5). After the sample was qualified, the rRNA was removed with the Ribo-Zero kit (Epicentre, San Diego, CA). Metatranscriptomic libraries were constructed according to the protocol of the NEBNext R Ultra RNA library prep kit (Illumina; New England BioLabs, Ipswich, MA) and were then sequenced on an Illumina HiSeqTM2500/4000 platform at Beijing Auwigene Tech, Ltd. (Beijing, China). Raw metatranscriptomic data were processed by removing the rRNA sequences and low-quality reads (Q < 0.02). Trinity was employed for *de novo* transcriptome assemblies (42). Prodigal (43) software was used to conduct open reading frame (ORF) prediction on the assembled contig sequence, and CDHIT (44) software was used to remove redundant genes from the predicted gene sequence with a similarity of 0.95 to obtain a non-redundant gene set. The subsequent analysis of gene expression levels was based on nonredundant gene sequences. Finally, databases such as EuKaryotic Orthologous Groups/Clusters of Orthologous Groups (KOG/COG) (https://www.ncbi.nlm.nih.gov/research/cog-project/), KEGG, Carbohydrate-Active Enzyme (CAZyme), eggNOG, and Comprehensive Antibiotic Resistance Database (CARD) were used to obtain functional annotation information.

**Network construction and visualization.** A network analysis was carried out to compare the microbial network structures of four batches. Construction of molecular ecological networks was carried out in an open-access pipeline (http://ieg4.rccc.ou.edu/mena/) based on a Random Matrix Theory as described previously (17, 45). Briefly, the whole process of Molecular Ecological Network Analysis Pipeline (MENA) included two phases, namely, network construction and network analyses. The network construction can be divided into four major steps, as follows: data collection, data transformation/standardization, pairwise similarity matrix calculation, and the adjacent matrix determination by Random Matrix Theory-based approach. The network analyses were composed of network topology characterization, module detection, module-based gene analysis, and identification of modular roles. The module separation was based on the fast greedy modularity optimization. Z-P plot was conducted based on parameters of modularity analysis (45). Key OTUs, including module hubs and connectors, were identified by values of $Zi$ and $Pi$. The threshold values of $Zi$ and $Pi$ for categorizing OTUs were 2.5 and 0.62, respectively.

**Statistical analysis.** Multivariate statistical analyses on volatile flavors were performed with the SIMCA-14.1 software (Umetricus, Sweden). Principal-component analysis (PCA) was used to investigate the favor data in different batches. One-way analysis of variance (ANOVA) with Duncan's test was employed to investigate statistical differences. Differences between groups with $P$ values of <0.05 were considered to be statistically significant. One-way ANOVA was conducted using OriginPro2021 software (OriginLab Corporation, MA, USA). The dynamics of microbial diversity and modularity across different batches were fitted via OriginPro2021. Pearson correlation coefficients were calculated based on linear fit, and $R^2$ goodness of fit values were calculated. The network was created by Gephi (v.0.9.2) to sort through and visualize the correlations between microbiota (46).

**Data availability.** The amplicon and transcriptomic databases were submitted to the NCBI Sequence Read Archive (SRA) and are available under accession number PRJNA828351 and PRJNA82834.

## SUPPLEMENTAL MATERIAL

Supplemental material is available online only.

**FIG S1**, TIF file, 0.3 MB.
**FIG S2**, TIF file, 1 MB.
**FIG S3**, TIF file, 0.5 MB.
**FIG S4**, TIF file, 1.3 MB.
**FIG S5**, TIF file, 0.7 MB.
**FIG S6**, TIF file, 1.8 MB.
**TABLE S1**, XLSX file, 0.02 MB.
**TABLE S2**, XLSX file, 0.01 MB.

## ACKNOWLEDGMENTS

This work was supported by the China Postdoctoral Science Foundation (2021M701469), the National Natural Science Foundation of China (NSFC) (grant 32172176), the Natural Science Foundation of the Jiangsu Province of China (grant

BK20201341), and the National First-class Discipline Program of Light Industry Technology and Engineering (LITE2018-12).

We thank Fei Hao, Xibin Lv, and Jianjun Lu for providing the statistical methods and for carrying out sample collection.

We declare no competing interests.

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
