## [Reviewer comments · mSystems]

Fungal interactions strengthen the diversity–functioning relationship of solid-state fermentation systems

Hongxia Zhang, Yuwei Tan, Junlin Wei, Hai Du, and Yan Xu

Corresponding Author(s): Yan Xu, Jiangnan University

Review Timeline:

Submission Date:	May 2, 2022
Editorial Decision:	June 7, 2022
Revision Received:	June 12, 2022
Accepted:	June 15, 2022

Editor: Stephen Lindemann

Reviewer(s): Disclosure of reviewer identity is with reference to reviewer comments included in decision letter(s). The following individuals involved in review of your submission have agreed to reveal their identity: Xueshan Wang (Reviewer #4)

Transaction Report:

DOI: <https://doi.org/10.1128/msystems.00401-22>

June 7, 2022

Prof. Yan Xu
Jiangnan University
Lihu avenue 1800
Wuxi
China

Re: mSystems00401-22 (Fungal interactions strengthen the diversity-functioning relationship of solid-state fermentation systems)

Dear Prof. Yan Xu:

Thank you for submitting your manuscript to mSystems. We have completed our review and I am pleased to inform you that, in principle, we expect to accept it for publication in mSystems. However, acceptance will not be final until you have adequately addressed the minor comments and suggestions of Reviewer 2.

Preparing Revision Guidelines

Sincerely,

Stephen Lindemann

Editor, mSystems

Journals Department
Reviewer comments:

Reviewer #1 (Comments for the Author):

I note that the authors made the suggested changes

Reviewer #4 (Comments for the Author):

The manuscript (mSystems00401-22) entitled "Fungal interactions strengthen the diversity-functioning relationship of solid-state fermentation systems" tries to identify the effect of diversity on microbial functions in a complex spontaneous food fermentation process. It is interesting to apply these advanced technologies to analyze the microbial communities in the traditional fermentation process. The manuscript is exceptionally well written, which I think can be regarded for publication. However, there are some minor issues that would require clarification or modification before deciding to accept this study for publication.

1. In "ABSTRACT" part, the essential elements of a good abstract are available, but the description of the results should be as accurate as possible. Line 21, the quality of the Baijiu seems to include all kinds of evaluations, including flavor components and sensory evaluation. In this paper, the quality of the Baijiu focused on the metabolic profiles, so the description should be revised.
2. Line 23, "Metabolomics analysis" in "ABSTRACT" part was not mentioned in "Results" part. Please clarify this concept clearly.
3. In "KEYWORDS" part, "Baijiu" should be replaced by "Baijiu".
4. In "INTRODUCTION" part, Microbial diversity and structure are important, in particular the fungal microbial group, is crucial for quality and productive food fermentation in a spontaneous process. Line 95, it is inappropriate to emphasize the importance of yeast only.
5. Figure 1 showed the metabolic profiles of sauce flavor Baijiu in different batches, the annotation for the heat map should be added in the figure.
6. Figures 2A and 5 should be in a unified format and changed into a closed frame.
7. The multi-batch Baijiu fermentation is a complex and important fermentation process. It is a good idea to use it as a model to study community diversity and function in this paper. The metabolic profiles and ethanol productivity of Baijiu are measurable functions of the Baijiu fermentation microbial community in terms of its macroscopic performance. Therefore, the title of result 1 should be reconsidered (Line 137).
8. Due to the complex physical and chemical components of fermented grains, which often contain sugar polyphenols and secondary metabolites of microorganisms, the sample treatment and experimental conditions of fermented grains are complicated. In this paper, it is valuable to construct three part metabolic pathways based on transcriptome, but we also hope to further explore this part of data: for example, starting with functional genes and functional microorganisms, which is of great significance to the analysis of liquor fermentation mechanism. Please strengthen this outlook in the discussion part.
9. In "Materials and methods" part, line 500 "0.25mm" should be replaced by "0.25 mm", line 519 "4°C" should be replaced by "4 °C".
10. In Fig. 7 "Pyruvate metabolism" part, "NADP+" should be replaced by "NADP+", "NAD(P)+" should be replaced by "NAD(P)+".
11. Line 511, the sentence seems to repeat the meaning of the following sentence. Please check and delete or modify it.
12. In "REFERENCE" part, line 664, 727, 741, 747, 749, and 753, "-" should be replaced by "-".

June 15, 2022

Prof. Yan Xu
Jiangnan University
Lihu avenue 1800
Wuxi
China

Re: mSystems00401-22R1 (Fungal interactions strengthen the diversity-functioning relationship of solid-state fermentation systems)

Dear Prof. Yan Xu:

Your manuscript has been accepted, and I am forwarding it to the ASM Journals Department for publication. For your reference, ASM Journals' address is given below. Before it can be scheduled for publication, your manuscript will be checked by the mSystems production staff to make sure that all elements meet the technical requirements for publication. They will contact you if anything needs to be revised before copyediting and production can begin. Otherwise, you will be notified when your proofs are ready to be viewed.

Publication Fees:

We recognize that the video files can become quite large, and so to avoid quality loss ASM suggests sending the video file via <https://www.wetransfer.com/>. When you have a final version of the video and the still ready to share, please send it to mSystems staff at mSystems@asmusa.org.

For mSystems research articles, if you would like to submit an image for consideration as the Featured Image for an issue, please contact mSystems staff at mSystems@asmusa.org.

Sincerely,

Stephen Lindemann
Editor, mSystems

Journals Department
Fig. S4: Accept
Fig. S3: Accept
Table S2: Accept
Fig. S6: Accept
Fig. S1: Accept
Fig. S2: Accept
Fig. S5: Accept
Table S1: Accept